# Hepatoprotective Activity of Ethanol Extract of Rice Solid-State Fermentation of *Ganoderma tsugae* against CCl_4_-Induced Acute Liver Injury in Mice

**DOI:** 10.3390/molecules27165347

**Published:** 2022-08-22

**Authors:** Xin Zhang, Wentao Lv, Yongping Fu, Yu Li, Jinhe Wang, Dongjie Chen, Xuerong Han, Zhenhao Li

**Affiliations:** 1Engineering Research Center of Chinese Ministry of Education for Edible and Medicinal Fungi, Jilin Agricultural University, Changchun 130118, China; 2Mudanjiang Branch of Heilongjiang Academy of Agricultural Sciences, Mudanjiang 157000, China; 3ShouXianGu Botanical Drug Institute, Hangzhou 311100, China

**Keywords:** *Ganoderma tsugae*, rice solid-state fermentation, response surface optimization, triterpenoids, hepatoprotection

## Abstract

*Ganoderma tsugae* is well known as a medicinal mushroom in China and many Asian countries, while its fermentation technique and corresponding pharmacological activity are rarely reported. In this study, a wild *G. tsugae* strain (G42) with high triterpenoid content was screened from nine strains by rice solid-state fermentation, and 53.86 mg/g triterpenoids could be produced under optimized conditions; that is, inoculation amount 20%, fermentation temperature 27 °C, and culture time 45 days. The hepatoprotective activity of G42 ethanol extract was evaluated by CCl_4_-induced liver injury in mice, in which changes in the levels of aspartate aminotransferase (AST), alanine aminotransferase (ALT), oxidation-related factors, and inflammatory cytokines in serum or liver samples demonstrated the therapeutic effect. In addition, the ethanol extract of G42 reduced the incidence of necrosis and inflammatory infiltration, and decreased protein expression levels of phosphor-nuclear factor-κB (NF-κB), interleukin-Iβ (IL-1β), and nuclear factor erythroid-2-related factor 2 (NRF2). The chemical composition of the ethanol extract was analyzed by high-resolution mass spectrometry and molecular networking. Three main triterpenoids, namely platycodigenin, cucurbitacin IIb, and ganolecidic acid B were identified. This work provided an optimized fermentation method for *G. tsugae*, and demonstrated that its fermentation extract might be developed as a functional food with a hepatoprotective effect.

## 1. Introduction

Dysfunction caused by liver damage due to alcohol, viruses, chemical toxins, unhealthy diet, and overwork is the primary inducement of liver cancer formation [1]. Therefore, the mitigation and functional recovery of liver injury are the means to prevent the further deterioration of liver diseases. The hepatoprotective activities of the family of Ganodermataceae have been extensively studied. *Ganoderma lucidum* capsules significantly reduced the fat content of livers in subjects with moderate fatty liver [2]. Water extract of *G. lucidum* showed a protective effect on hepatic damage by improving oxidative stress and restoring the mitochondrial enzyme activities and membrane potential [3]. Ethanol extract of *G. lucidum* also relieved alcohol-induced liver injury in mice by inhibiting lipid peroxidation and inflammatory immune response [4]. It is also reported that the crude extract of *G. tsugae* can alleviate the degree of liver fibrosis in mice with CCl_4_-induced liver injury [5]. However, the active ingredients responsible for the hepatoprotective effects and underlying mechanisms of *G. tsugae* remain elusive.

The fruiting body of *G. tsugae* contain a variety of phytochemicals, such as polysaccharides, triterpenoids, sterols, anthraquinones, nucleosides, and trace elements. Generally, polysaccharides and triterpenoids are considered as active components [6]. Water-soluble polysaccharides of the fruiting body are mainly composed of mannose, glucose, and galactose linked by β-glycosidic bonds with different molecular weights [7], and have lipid-lowering and hepatoprotective activities. Nine unique triterpenoids isolated from the fruiting body of *G. tsugae* consists of six lanostane triterpenoids (tsugaric acid C(1), D(2), E(3) and F(4), tsugarioside B(5), and C(6)) and three seco-lanostanoids. Among lanostane triterpenoids, (1), (5), and (6) had inhibitory activities on the growth of liver cancer cells, and (2), (3), and (4) could inhibit xanthine oxidase (XO), while three seco-lanostanoids showed weak toxicity to PC3 cell line [8,9,10]. However, considering the long cultivation period and low yields of secondary metabolites, extracting of active ingredients from the fruiting body for structural elucidation and pharmacological studies is relatively time-consuming and resource-intensive. A more efficient way to produce these active ingredients is much sought after.

Recently, fermentation mycelia of edible and medicinal mushrooms were employed as a new source for extraction of active ingredients substituting the fruiting bodies. It has been reported that the polysaccharide content of liquid fermentation mycelium of *G. tsugae* was significantly higher than the fruiting body [11]. Triterpenoid content obtained from *G. tsugae* cultured by a solid medium of potato glucose agar (PDA) has a 16.1-fold enhancement compared to liquid fermentation, and exhibited stronger inhibitory effect against human lung cancer cells CH27 [12]. The solid-state fermentation (SSF) of *G. lucidum* using soybean as culture substrate produces ergosterol content 30% higher than that of the fruiting body [13]. Six kinds of phenolics were extracted and identified from *Ganoderma lipsiense* grown in red rice medium using SSF techniques [14,15]. A new compound sesquiterpene and six isolactarane-related norsesquiterpenes were isolated from the solid culture of *Flammulina velutipes* [16]. All these studies have shown that SSF shares advantages with liquid fermentation such as short culture cycle, controllable culture conditions, and easy scale-up production, while also exhibiting diversity and novelty of chemical compositions in the fermentation products [17]. Until now, however, research on the SSF method for *G. tsugae* has been limited, as are data on the active ingredients and pharmacological effects.

In this paper, a new SSF method of *G. tsugae* with cooked rice as the only solid substrate was developed, and a wild strain of *G. tsugae* (G42) with high triterpenoid content was screened from nine strains. The fermentation method was optimized in terms of inoculation amount, fermentation temperature, and culture time by response surface methodology, to obtain high triterpenoid yield. The hepatoprotective activity of ethanol extract of G42 was evaluated by analyzing superoxide dismutase (SOD), glutathione peroxidasel (GSH), and malondialdehyde (MDA), as well as inflammatory factors interleukin-2 (IL-2), interleukin-8 (IL-8), interleukin-10 (IL-10), tumor necrosis factor-α (TNF-α), Interferon-β (IFN-β), and interleukin-1β (IL-1β) levels in a CCl_4_-induced acute liver injury mouse model. The main triterpenoids of the ethanol extract were identified by high-resolution mass spectrometry and molecular networking.

## 2. Results

### 2.1. Screening of Optimal G. tsugae Strain with High Triterpenoid Content

According to the results of the team’s previous experiments, it was found that these nine Strains of *G. tsugae* have genetic diversity, so further experiments were conducted on them. To obtain *G. tsugae* with high triterpenoid content, rice was used for screening the strain as a solid substrate. The biomass and triterpenoid content of nine strains of *G. tsugae* from different sources were compared with *G. sichuanense*. The results showed that the biomass of different strains was 166.11–188.76 g, and the triterpenoid content was 19.73–48.94 mg/g. Among them, the biomass of G42 and G43 were 187.78 g and 188.76 g, respectively, which were higher than *G. sichuanense* and other strains. The triterpenoid content of G42 was 48.94 mg/g, which was significantly higher than that of other strains (Table 1). DPPH clearance rate and total reducing ability were determined to verify the antioxidant activity of the ethanol extracts of fermentation products. The results showed that the total reducing capacity and DPPH clearance rate of G42 were 1.24 and 90.96%, respectively, which were higher than other strains (Table 1). G42 was selected as the high-yield triterpenoids strain for subsequent experiments based on the above results.

### 2.2. Optimization of Culture Conditions of G42

#### 2.2.1. Effects of Single-Factor on Triterpenoid Content of G42

Using the content of triterpenoids in the fermentation as an indicator, when the inoculum volume was 20 mL and the fermentation time was 40 d, the mycelium of *G. tsugae* could accumulate triterpenoids in the temperature range of 20–30 °C, and the amount was up to 51.27 mg/g at 27 °C (Appendix A). When the inoculum volume was 20 mL and the fermentation time was 45 d, the accumulated triterpenoids reached the maximum of 50.63 mg/g (Appendix A) at 25 °C, and the content of triterpenoids decreased to 48.75 mg/g after the fermentation time continued to extend. When the fermentation temperature was 25 °C, the culture time was 40 d, and the inoculum volume was 25 mL, the triterpenoid content reached the highest level (Appendix A).

#### 2.2.2. Optimization of Culture Conditions of G42 by Response Surface Methodology

The response surfaces methodology can solve multivariate problems by analyzing regression equations to find the optimal process parameters. According to the single factor results, the temperature of 20, 25, and 30 °C were selected as the factor A; the fermentation time of 40 d, 45 d, and 50 d were used as the factor B; and the inoculum volume of 20, 25, and 30 mL were the factor C (Appendix A). The equation model of triterpenoid content was obtained according to the fitting analysis of Design Expert 8.0.6 software, Y = 52.67 + 1.87 × A − 0.84 × B − 4.10C + 0.12A × B − 0.43 × A × C + 0.25 × B × C − 2.96 × A^2^ − 3.42 × B^2^ − 6.27 × C^2^, and the absolute value order of the first-order coefficient was C > A > B, indicating that the content of triterpenoids was most affected by the inoculum amount, followed by the culture temperature, and the least by the fermentation time. The model of response surface and the variance significance analysis of each coefficient of the regression equation can show the model’s accuracy. The analysis results showed (Table 2) that the variance of the test model was significant (the smaller the *p*-value, the higher the model accuracy), and the data analysis had high reliability. In the model, the variance *p* of A, C, A^2^, B^2^, and C^2^ are all <0.0001, and the error of the calculation result of this equation was small. The 2D and 3D graphs of the response surface more intuitively showed that the inoculum amount and culture temperature had a greater impact on the triterpenoid content (Figure 1). According to the optimal combination of conditions given by the model, at 27 °C, fermentation time of 44 days, and inoculation amount of 23 mL, the content of triterpenoids was 53.86 mg/g, which was close to the predicted value of 53.98 mg/g given by the model.

### 2.3. Evaluation of the Hepatoprotective Activity of the Ethanol Extract

#### 2.3.1. Effects of the Ethanol Extract on Serum ALT and AST Activities

ALT and AST are landmark indicators for evaluating liver injury [18]. Compared with the CG, serum ALT, and AST levels in the CCl_4_ group were abnormally increased (*p* < 0.05, Figure 2A), indicating that CCl_4_ causes liver injury. PG significantly decreased the levels of ALT and AST in mice with liver injury (*p* < 0.05). The HG showed a more significant reduction in ALT and AST activity in serum compared to the other groups. These results indicate that the ethanol extract of G42 can resist elevated serum ALT and AST levels caused by CCl_4_-induced acute liver injury in mice.

#### 2.3.2. Effect of the Ethanol Extract on the Antioxidant Enzyme Activity of Liver Tissue

SOD and GSH of mouse liver tissue reflecting the free radical scavenging capacity and antioxidant capacity of cells were measured [19]. The activities of SOD and GSH showed the same trend (Figure 2B,C). The SOD and GSH activities of the CCl_4_ group were significantly decreased (*p* < 0.05), and the LG, MG, and HG showed a dose-dependent manner, and the HG was significantly higher than those in the PG. The results of MDA representing the potential antioxidant capacity showed that CCl_4_ treatment led to lipid peroxidation, and the level of MDA was significantly increased (Figure 2D). PG, LG, MG, and HG groups downregulated the level of MDA. The HG downregulated MDA close to the CG and was significantly better than the PG (*p* < 0.05). These results suggest that the ethanol extract of G42 can effectively alleviate oxidative stress levels caused by the destruction of the antioxidant defense systems.

#### 2.3.3. Histopathological Analysis of Liver and Kidney

The cell nuclei from livers (Figure 3A) and kidneys (Figure 3B) in the CCl_4_ group were absent and disordered, with poor intercellular boundaries and extensive cell necrosis (red arrows indicate areas of extensive necrosis) compared with those in the CG group. The PG showed a decrease in the number of necrotic tissue cells and an increase in the number of cells with intact nuclei, although nuclear cohesion was still observed. The LG, MG, and HG showed a gradual decrease in the area of necrotic cells with increasing dose concentrations, and the HG sharply reduced the number of necrotic cells and recovered liver tissue histology and renal histology.

#### 2.3.4. Effects of the Ethanol Extract on Inflammatory Factors of Liver Tissue

Inflammatory cytokines are involved in the inflammatory response and can respond to cell and tissue damage. The recovery of injured cells and tissues was evaluated by measuring the levels of inflammatory cytokines in liver-injured mice (Figure 4). Compared with the GC, the levels of IL-2, IL-8, IL-10, TNF-α, IFN-β, and IL-1β of liver tissue in the CCl_4_ group were significantly increased (*p* < 0.05), and the administration group decreased the level of all inflammatory factors in cells, the HG group had the most significant inhibitory the level of inflammatory factors. The IL-10, IFN-β, and IL-1β levels showed a clear dose-dependent effect (Figure 4A), while the IL-2 and TNF-α levels (Figure 4B) were not significantly changed.

#### 2.3.5. Effect of the Ethanol Extract on the IHC of Liver Tissue

NRF2 and NF-κB are key factors for regulating inflammatory and immune responses, and these two transcription factors restrict each other to maintain the intracellular redox balance [20,21,22]. To analyze the modulating effects of the ethanol extract of G42 on two transcription factors in liver-injured mice, the expression levels of p-NF-κB P65, IL-1β, and NRF2 in mice liver tissues were investigated (Figure 5A). IHC results showed that the IL-1β, NRF2, and p-NF-κB P65 in the CCl_4_ group had yellow-brown color compared with the GC, indicating that the liver tissue produced a severe inflammatory response. On the other hand, the LG, MG, HG, and PG could alleviate the inflammatory reaction symptoms. Quantitative analysis results showed (Figure 5B) that the LG, MG, HG, and PG could reduce the expression levels of the three proteins, and the triterpenoids administration group showed a dose-dependent manner. The HG was better than the PG (*p* < 0.05). These results indicated that CCl_4_ prompts the NF-κB signaling pathway, leading to an increase in the expression of p-NF-κB P65 and the transcription level of the inflammatory factor IL-1β. At the same time, activation of NF-κB signaling pathway inhibited the anti-inflammatory activity of NRF2, causing an inflammatory response, resulting in high expression of NRF2 in mice with liver injury.

### 2.4. Chemical Identification of the Ethanol Extract of G42

To further identify the potential active components that exerted hepatoprotective activity in the ethanol extract of G42, the extract was analyzed by UPLC–Q–TOF/MS. The chromatograms of the ethanol extract of rice (Figure 6A) and G42 (Figure 6B) show certain differences, indicating changes in composition after fermentation. Significantly varied chemical components were first screened by combining volcano map (Appendix A) and VIP value (Appendix A) by PCA. Then, the molecular network based on MS^2^ spectra similarity was created on GNPS to identify the differential components. The molecular network contained 367 distinguishable precursor ions, visualized as nodes in the network with 31 clusters (node ≥ 2) and 130 single nodes. In the molecular network, each node represents a compound, in which red represents the ethanol extract rice, blue represents the ethanol extract of G42, and the yellow circle represents identified differential components (Figure 6D). The area of different colors represents the relative content of the node compound in different samples. Fifty-six different components were identified by molecular networking analysis and UNIFI 1.9. Among them, platycodigenin, cucurbitacin IIb, and ganolucidic acid B were the triterpenoids only detected in the ethanol extract of G42 while not in rice (Figure 6C). Other components were phospholipids also contained in rice. Detailed MS information of differential components is displayed in Appendix A.

## 3. Discussion

In this study, we developed a new SSF method for *G. tsugae* using rice as the sole substrate, which is reported for the first time. This method screened out a new *G. tsugae* strain G42 with a high yield of triterpenoids (53.86 mg/g) and antioxidant activity. Wang et al. have reported that two novel sterpurane sesquiterpenes with antibacterial activity were isolated from the SSF of *F. velutipes* [23]. Cooked rice as a fermentation substrate not only provides nutrition similar to an artificial cultivation environment for fungi, but also affect metabolic pathways of secondary products of fungi [24]. Compared with the use of fruiting bodies of *G. tsugae* artificially cultivated for extracting of pharmacological active ingredients, SSF had advantages of controllable culture conditions, short culture cycle, low labor cost. Importantly, safety is a critical concern in artificially cultivation of *G. tsugae*, since the covered soil may introduce exogenous contaminants [25]. In contrast, the procedure of SSF is more controllable and minimizes contamination. Therefore, SSF is an important supplementary culture method for extracting or synthesizing active ingredients, due its short cycle, high levels of active substances, and high safety.

The sharp increase in levels of ALT and AST in serum of CCl_4_-induced mice showed that the liver was damaged by the free radicals produced from CCl_4_ [26]. The ethanol extract of G42 could increase SOD and GSH activity, and inhibit the increase of the MDA content in the CCl_4_-treated mice. These results suggest that the ethanol extract of G42 plays a critical role in salvaging free-radical activities and lowering oxidative stress [27]. In addition, the ethanol extract of G42 shows anti-inflammatory properties by reducing the levels of inflammatory factors IL-2, IL-8, IL-10, TNF-α, IFN-β, and IL-1β in liver-injured mice, which is similar to the ethanol extract of *G. lucidum* [4]. Moreover, NF-κB is a central transcription mediator that regulates transcription levels of inflammatory cytokines. The suppression of NF-κB activation greatly decreases the levels of proinflammatory cytokines [28]. The ethanol extract of G42 effectively inhibited the NF-κB signaling pathway activated by the CCl_4_, thereby suppressing the transcription level of inflammatory factor IL-1β. NRF2 activation is a major regulator of the antioxidant response and enhances the activity of the innate immune system, thereby attenuating or eliminating the effects of toxic and harmful substances on organisms [29]. Thus, the ethanol extract of G42 could hepatoprotective by regulating NF-κB/NRF2-related oxidative stress and inflammatory pathways. These effects are similar to the triterpenoids and polysaccharides from *G. lucidum* or *G. applanatum* [30,31,32]. All data suggested that the synergistic regulation of NF-κB/NRF2 signaling factors (generally induced by the innate immune system) plays a central role in the hepatoprotective effect against acute CCl_4_-induced liver injury. The study provides evidence to support the use of SSF of rice with *G. tsugae* as a natural functional product that can protect against liver injury.

A total of 56 differential components were identified between the SSF of G42 and blank rice by PCA and molecular networking. Among them, three triterpenoids, ganolucidic acid B, platycodigenin, and cucurbitacin IIb were only found in G42. Ganolucidic acid B is a unique triterpenoid to the genus *Ganoderma*, and has been reported with cytotoxic activity against the human cancer cell line HepG2 [33,34]. Platycodigenin and Cucurbitacin IIb are the main active components of the medicinal plants *Platycodon grandiflorum* and Cucurbitaceae plants, respectively, and are known to possess the ability to downregulate the activities of pro-inflammatory factors IL-1β, TNF-α, and IL-6 [35,36]. Above all, the ethanol extract of G42 may exert its hepatoprotective activity mainly through three triterpenoids. The type of triterpenoids compounds isolated from the ethanol extract of G42 are less than the fruiting bodies may be due to the effect of the rice matrix on secondary metabolites produced. It is necessary to further expand the scale of rice SSF and further analyze the types of triterpenoids in fermentation products. Moreover, the mechanism of their hepatoprotective effects of regulate/inhibit the level of NF-κB through DNA binding and p65 translocation need to be investigated.

So far, we have integrated the experimental and literature evidence to prove that G42 ethanol extract could partially alleviate the CCl_4_-induced liver injury by reducing oxidative stress and inflammatory pathways. To systematically investigate the effects of G42, we would like to employ the structural activity relationship model and molecular docking to predict potential bioactivity and targets for G42 components based upon identified structures and reported activities in our future study. Followed by such studies, experimental validation would reveal comprehensive effects of G42.

## 4. Materials and Methods

### 4.1. Strains and Reagents

Four cultivated *G. tsugae* strains, five wild *G. tsugae* strains, and one *G. sichuanense* strain were selected for SSF. These strains (Appendix A) were stored in the Engineering Research Center of Edible and Medicinal Fungi, Ministry ofEducation, Jilin Agricultural University (Changchun, China).

Maltose, glucose, absolute ethanol, perchloric acid, vanillin, ethyl acetate, concentrated sulfuric acid, phenol, ether, carbon tetrachloride (CCl_4_), and xylene were purchased from Sinopharm Chemical Reagent Co., Ltd. (Shanghai, China). Oleanolic acid standard, Hematoxylin-eosin stain, and 4% paraformaldehyde solution were ordered from Solarbio (Beijing, China). Alanine aminotransferase (ALT) and Aspartate aminotransferase (AST) kits were purchased from Jiangsu Jingmei Biotechnology Co., Ltd. (Yancheng, China). Malondialdehyde (MDA), Glutathione (GSH), and Superoxide dismutase (SOD) kits, and Mouse IL-2, IL-8, IL-10, TNF-α, IFN-β, and IL-1β enzyme-linked immunosorbent assay (ELISA) kits were purchased from Nanjing Jiancheng Bioengineering Institute (Nanjing, China). PBS buffer was purchased from Dingguo Changsheng Biotechnology Co., Ltd. (Beijing, China), Silybin Capsules were purchased from Tasly Pharmaceutical Group Co., Ltd. (Tianjin, China), and NRF2 Rabbit pAb, IL-Iβ Rabbit pAb, p-NF-κB P65 Rabbit pAb, and horseradish peroxidase-conjugated goat anti-rabbit antibodies were purchased from ABclonal (Wuhan, China).

### 4.2. Screening of High-Yield Triterpenoids Strains of G. tsugae from Rice by SSF

#### 4.2.1. SSF Culture of *G. tsugae* Strains

Mycelium was inoculated into a 250 mL conical flask containing 100 mL of medium (0.4% glucose, 0.8% maltose, and 0.4% yeast powder), and cultured at 25 °C and 150 rpm for 10 days to obtain a seed liquid. A volume of 20 mL of seed liquid was inoculated into the rice medium and cultured at 25 °C for 40 d. Finally, fermentation products were collected, and mycelium biomass was calculated.

#### 4.2.2. Triterpenoid Content in Fermentation Products

After drying, the fermentation products of each strain were ground into powder and passed through a 200-mesh sieve. An amount of 10 g of the fermentation products was taken, and 200 mL 70% ethanol (solid-liquid ratio 1:20) was added to a round bottom flask, and then extracted with hot water at 80 °C for 100 min. Finally, the supernatant was collected after centrifuged at 8000 rpm/min for 10 min. Vanillin–glacial acetic acid and perchloric acid colorimetric spectrophotometry were used to determine the triterpenoid content [37].

#### 4.2.3. Antioxidant Activity In Vitro of Ethanol Extract of Fermentation Products

The ethanol extract was prepared according to Section 4.2.2. The DPPH clearance rate and total reducing ability were determined as described by Zhang et al. [38].

### 4.3. Optimization Conditions of G42 with High Triterpenoid Content by Response Surface Methodology

#### 4.3.1. Effects of Culture Conditions on Triterpenoid Content

Temperature: 20 mL seed culture medium was inoculated in sterilized glass bottles (500 mL) containing 160 g rice medium (5 bottles) and placed in incubators at 20, 23, 25, 27, and 30 °C, respectively. After 40 days of culture, the content of triterpenoids was determined according to Section 4.2.2.

Time: The culture temperature was 25 °C, the inoculum volume was 20 mL, and the culture days were set as 30, 35, 40, 45, and 50 d, respectively, and the post-processing and determination methods were the same as above.

Inoculation volume: Control the culture time for 40 d, the temperature at 25 °C, and the inoculation volume as 10, 15, 20, 25, and 30 mL, respectively, and the post-processing and determination methods were the same as above.

#### 4.3.2. Response Surface Design of G42 from Rice by SSF

The single factor test results selected the parameter range of the response surface test. Temperature (A), fermentation time (B), and inoculum amount (C) as variables, and using triterpenoid content as response value, the optimal process parameters for SSF of rice with G42 were obtained by response surface analysis. Design Expert 8.0.6 was used to design and analyze the response surface of the obtained data

### 4.4. Evaluation of the Hepatoprotective Activity of Ethanol Extract of G42

#### 4.4.1. Preparation of G42 Extracts

The ethanol extract was obtained from G42 fermentation products under optimized conditions. The ethanol extracts were freeze-dried and weighed after removing ethanol using a FreeZone vacuum evaporator. The dried product was dissolved in H_2_O and sterilized by a 0.2-μm filter for in vivo study.

#### 4.4.2. Experimental Design of Animals

Experimental mice: Adult male ICR mice (body weight 20 ± 2 g) were purchased from Liaoning Changsheng Biotechnology Co., Ltd. (Liaoning, China).

Feeding conditions: Animal room temperature was 25 ± 1 °C, relative humidity was 40 ± 10%, 12 h light/12 h dark cycle. After seven days of adaptive feeding (food and water were free), 30 mice were randomly divided into six groups of five mice: control group (normal feeding, CG, n = 5), CCl_4_ model group (normal feeding, CCl_4_, n = 5), positive control group (73 mg/kg silymarin, PG, n = 5), Low-dose group (10 mg/kg, LG), Middle-dose group (20 mg/kg, MG), and High-dose groups (40 mg/kg, HG, n = 5). After seven days of feeding, except for the CG, all groups were intraperitoneally injected (i.p.) with 10 mL/kg 0.18% CCl_4_ soybean oil solution to induce acute liver injury for evaluating the hepatoprotective activity of the ethanol extract of G42.

Sampling: After the CCl_4_ soybean oil solution was treated (i.p.) for 24 h, blood samples were centrifuged at 5000 rpm for 10 min at 4 °C to collect serum. PBS was added to liver tissue, and a homogenate was prepared on ice. The homogenate was centrifuged at 5000 rpm for 20 min at 4 °C, and the supernatant was collected. The separated liver and spleen tissues were fixed in 10% neutral formalin for histological analysis. All procedures were approved by the Institution of Animal Care and Use of Jilin Agricultural University.

#### 4.4.3. Determination of Serum ALT and AST Levels

Serum ALT and AST levels were measured using Jiangsu Jingmei Institute kits following the manufacturer’s instructions.

#### 4.4.4. Inflammatory Cytokines and Antioxidant Activity of Liver Tissue

The levels of IL-2, IL-8, IL-10, TNF-α, IFN-β, and IL-1β in liver tissue were measured using ELISA kits from Nanjing Jiancheng Bioengineering Institute according to the instructions; MDA, SOD, and GSH levels in liver tissue were measured using Nanjing Jiancheng Bioengineering Institute kits following the manufacturer’s instructions.

#### 4.4.5. Histopathological Analysis

Liver and spleen tissues fixed in formalin were dehydrated, then degreased in xylene, paraffin-embedded, and cut into 5 µm slices. Sections were examined using hematoxylin-eosin (H&E) staining for histopathological response to liver injury.

#### 4.4.6. Immunohistochemistry (IHC)

Antigen retrieval was carried out by incubation in saline-sodium citrate buffer (pH 6.0) via autoclaving. After being washed with PBS, the sections were subjected to endogenous peroxidase blocking with 3% H_2_O_2_ in dark for 30 min. Then slices were washed with PBS and added primary antibody (NRF2 Rabbit pAb, IL-Iβ Rabbit pAb, and p-NF-κB P65 Rabbit mAb4) overnight at 4 °C. The slices were washed with PBS three times and then incubated HRP Goat Anti-Rabbit IgG at room temperature for 45 min. In the end, DAB color solution was dropped to observe the color state under a microscope. The sections were re-stained with hematoxylin staining solution for 2 min, washed with deionized water, differentiated with 0.5% alcohol hydrochloride for 4 s, and rinsed with running water to make the sections turn blue. Finally, the slices were dehydrated, transparent, and sealed with neutral gum. After drying, slices were observed and photographed under a microscope.

### 4.5. Chemical Identification of the Ethanol Extract of G42 by Mass Spectrometry

#### 4.5.1. Liquid Chromatography and Mass Spectrometry (LC-MS)

An ACQUITY I-Class UPLC system (Waters, Milford, MA, USA) coupled with a SYNAPT XS Q-TOF MS (Waters, Milford, MA, USA) was employed for the structural identification of triterpenoids in the ethanol extract of G42. Samples were prepared by 2.4.1, the rice was used as a control. The injection volume of samples was 1 µL. Chromatographic separation was carried out by an Acquity HSS T3 column (2.1 mm × 100 mm, 1.8 μm, Waters) at 25 °C with mobile phase A (0.1% *v*/*v* formic acid-water) and mobile phase B (acetonitrile). The flow rate was 0.45 mL/min, and a linear gradient elution was programmed: 0 min, 20% B; 0–2 min, 20–26.5% B; 2–9 min, 26.5% B; 9–19 min, 26.5–35% B; 19–28 min, 35–60% B; 28–32 min, 60–70% B; 32–37 min, 70–90% B; 37–40 min, 90–100% B; 40–45 min, 100% B.

Analysis of ingredients separated by UPLC was performed by SYNAPT XS Q-TOF MS in negative electrospray ionization (ESI-) mode under the following parameters: scan range, *m/z* 50–1200; collision energy, 25–50 eV; source temperature, 140 °C; cone voltage, 35 V; capillary voltage, 2.5 kV. Nitrogen was used as the desolvation gas. The desolvation gas was at 1000 L/h, with a temperature of 450 °C and a gas pressure of 6 bar.

#### 4.5.2. Data Processing and Compound Identification

Raw MS data processing was first conducted by the Progenesis QI (Waters, Milford, MA, USA) software in two respects: (1) peak alignment, baseline correction, calculation of information content, and deconvolution of overlapping peaks; and (2) data pretreatment of the MSE data for multivariate statistical analysis, achieving the conversion of 3D LC-MS raw data into a 2D matrix with the Retention Time-Exact Mass (RTEM) pair and peak intensity as the variable and observation ID. Subsequent multivariate statistical analysis of the above data was performed by SIMCA-P 14.1 in terms of principal component analysis (PCA), which screened chemical components varied significantly after fermentation.

The differential chemical components were identified by constructing MS/MS molecular networking and UNIFI 1.9 (Waters, USA). The MS/MS molecular network was generated using the GNPS platform. The MSE data were first converted to a CSV file and an MSP file using Progenesis QI [39] and then uploaded to GNPS to create the molecular networks. The precursor ion mass tolerance was set to 0.5 Da and the production tolerance to 0.1 Da. A network was constructed using three minimum matched peaks and a cosine score above 0.7. The spectra in the network were searched against the spectral libraries on GNPS to identify the components. Results were open and visualized using the Cytoscape 3.7.2.

Besides GNPS, component identification was also performed by UNIFI 1.9. The MSE data of the samples were imported into UNIFI 1.9. The compound was then automatically identified in terms of molecular weight and MS/MS fragmentation with the libraries of UNIFI. A margin of molecular weight error up to 5 ppm was allowed.

## 5. Conclusions

A wild *G. tsugae* strain G42 with high content of triterpenoids was screened with rice as the solid fermentation substrate. The strain G42 could produce 53.86 mg/g of triterpenoids using SSF of rice at 27 °C, 45 days, and an inoculum of 20%. The main active triterpenoids of ethanol extract G42 are platycodigenin, cucurbitacin IIb, and ganolecidic acid B. The ethanol extract G42 can significantly improve NRF2/NF-κB-related oxidative stress and inflammatory response, which is characterized by the increase of antioxidant enzyme activity and the decrease the levels of inflammatory factors IL-2, IL-8, IL-10, TNF-α, IFN-β, and IL-1β. This work provided an optimized fermentation method for *G. tsugae*, as well as the basis for the further development of functional food with hepatoprotective effect.

## Figures and Tables

**Figure 1 molecules-27-05347-f001:**
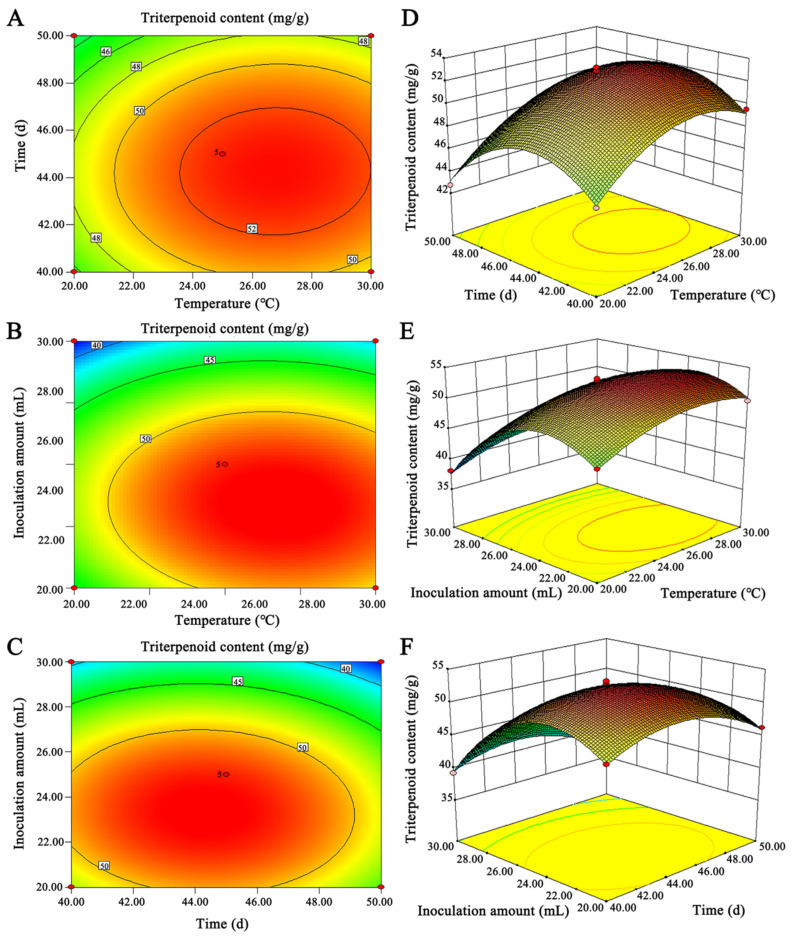
Response surface and contour map of the influence of the interaction of various factors on the content of triterpenoids. (**A**–**C**) Two-factor interactive contour maps and (**D**–**F**) two-factor interactive 3D maps.

**Figure 2 molecules-27-05347-f002:**
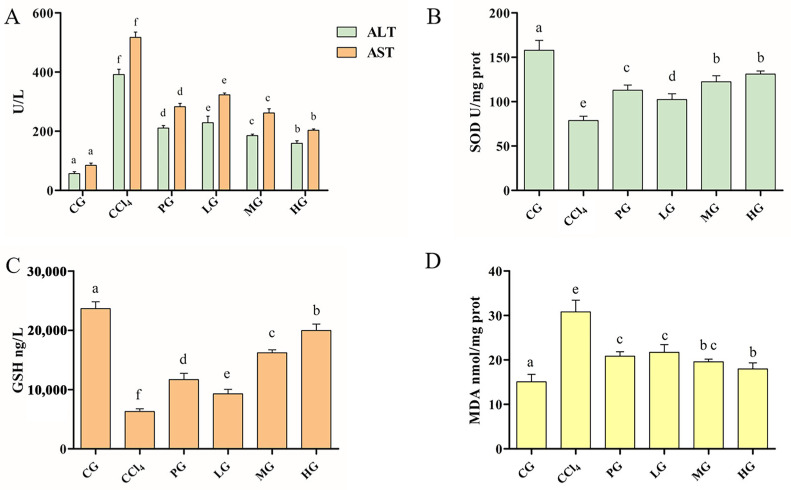
Effect of the ethanol extract of G42 on serum marker enzymes and oxidative stress level in the liver of acute liver injury caused by CCl_4_. (**A**) ALT and AST and (**B**) SOD level. (**C**) GSH and (**D**) MDA. The results are shown as mean ± SD. Different letters above each bar in the same parameter indicate a significant difference (*p* < 0.05) (*n* = 5).

**Figure 3 molecules-27-05347-f003:**
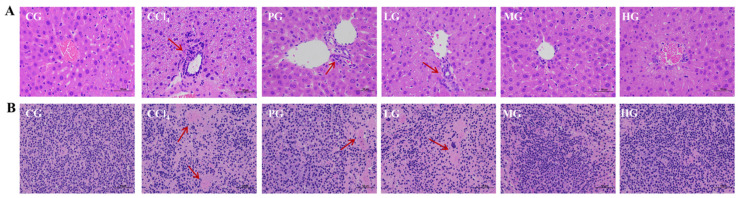
Effect of the ethanol extract of G42 on histopathological changes of liver and spleen caused by CCl_4_. (**A**) Liver and (**B**) spleen. (Scale bar: 50 μm; magnification: 400×).

**Figure 4 molecules-27-05347-f004:**
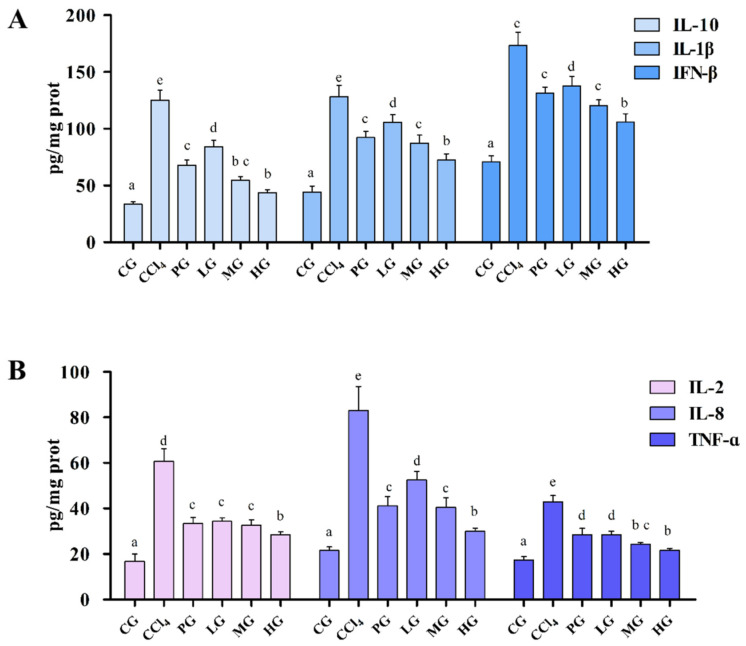
Effect of the ethanol extract of G42 on production of pro-inflammatory cytokines in liver determined using ELISA. (**A**) IL-10, IL-1β, and IFN-β levels in liver tissues, (**B**) IL-2, IL-8, and TNF-α levels in liver tissues. Different letters above each bar in the same parameter indicate the significant difference (*p* < 0.05) (*n* = 5).

**Figure 5 molecules-27-05347-f005:**
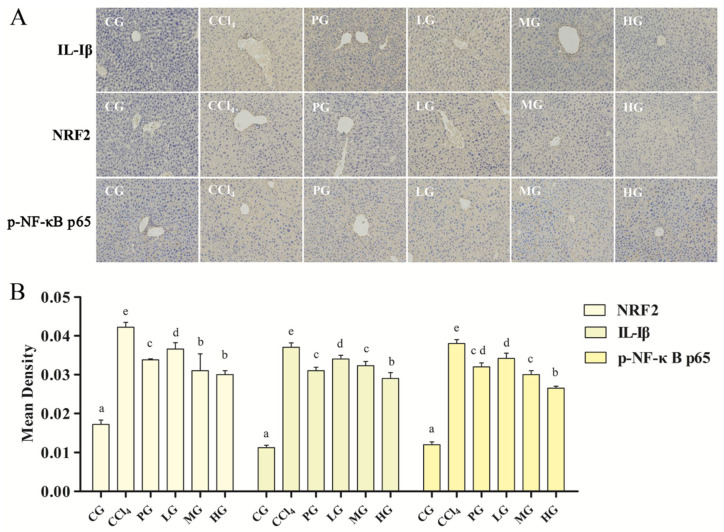
(**A**) Immunohistochemical results of hepatic IL-Iβ, NRF2, p-NF-κB p65 in acute liver injury mice (magnification 200×); (**B**) Quantitative analysis results, the same letter indicates no significant difference between groups (*p* > 0.05); the different letters indicate significant difference between groups (*p* < 0.05).

**Figure 6 molecules-27-05347-f006:**
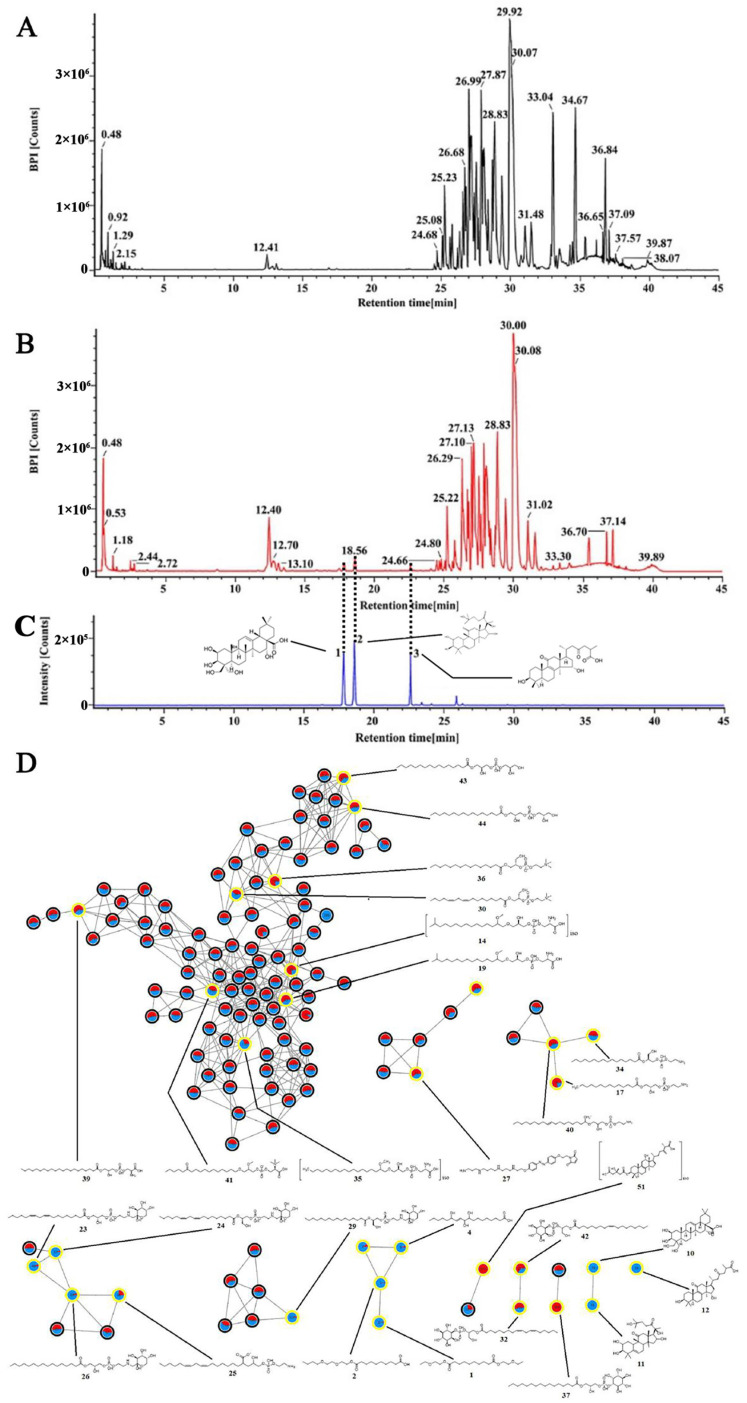
OUPLC–Q–TOF/MS chromatograms and annotation of the molecular networking. (**A**) OUPLC–Q–TOF/MS chromatograms of rice. (**B**) UPLC–Q–TOF/MS chromatograms of G42. (**C**) Three unique triterpenoids components in G42. The compound 1 is platycodigenin, reserve time is 17.80 ± 0.1, the molecular weight is C_30_H_48_O_7_, and the molecular weight is 520.3400. The compound 2 is cucurbitacin IIb, reserve time is 18.56 ± 0.1, the molecular weight is C_30_H_48_O_7_, and the molecular weight is 520.3400. The compound 3 is ganolucidic acid B, reserve time is 22.59 ± 0.1, the molecular weight is C_30_H_46_O_6_, and the molecular weight is 502.3294. (**D**) Molecular network of the tested samples, in which red represents components of the rice extract, blue represents components of the G42 extract, and yellow represents significantly varied components. The area of different colors represents the relative content of node components in different samples.

**Table 1 molecules-27-05347-t001:** Evaluation of SSF of *G.*
*tsugae* and *G. sichuanense*.

Number	Biomass (g)	Triterpenoid Content (mg/g)	Total Reducing Power	DPPH Clearance Rate (%)
G10	166.11 ± 1.23	31.79 ± 1.25	0.91	72.91
G12	169.54 ± 2.21	30.17 ± 2.03	0.28	45.89
G17	169.37 ± 1.57	19.85 ± 1.74	0.77	84.89
G20	171.2 ± 3.45	19.43 ± 1.65	0.40	54.75
G27	171.89 ± 2.36	35.03 ± 1.76	0.85	65.48
G40	174.86 ± 4.35	24.70 ± 2.14	0.55	65.11
G41	167.38 ± 3.66	34.15 ± 0.96	1.08	83.13
G42	187.78 ± 2.96	48.94 ± 2.26	1.24	90.96
G43	188.76 ± 2.39	27.94 ± 1.35	0.63	67.45
G78	177.69 ± 2.47	43.11 ± 1.39	0.79	79.02

**Table 2 molecules-27-05347-t002:** Analysis of variance of response surface test data results of content of triterpenoids.

Sum of Squares	df	Mean Square	F Value	*p*-Value Prob > F	Sig.
490.44	9	54.49	196.39	<0.0001	**
28.39	1	28.39	102.31	<0.0001	**
10.51	1	10.51	37.88	0.0005	*
154.88	1	154.88	558.18	<0.0001	**
0.060	1	0.060	0.22	0.6560	
0.64	1	0.64	2.31	0.1726	
0.56	1	0.56	2.03	0.1975	
29.24	1	29.24	105.38	<0.0001	**
58.90	1	58.90	212.28	<0.0001	**
181.35	1	181.35	653.56	<0.0001	**
1.94	7	0.28			
1.03	3	0.34	1.49	0.3446	-
0.92	4	0.23			
492.39	16				

Note: ** *p* < 0.01, * *p* < 0.05, - *p* > 0.05.

## Data Availability

Data generated or analyzed during this study are included in this published article and its Appendix A.

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
