# Peer review of "Hepatoprotective Activity of Ethanol Extract of Rice Solid-State Fermentation of Ganoderma tsugae against CCl4-Induced Acute Liver Injury in Mice"

_molecules, 2022, doi:10.3390/molecules27165347_

Round 1

Reviewer 1 Report

Please see in the attached file

Author Response

Response to Reviewer 1 Comments

Thank you for your comments. Our answers to your points are as follows.

Point 1: In general, there is a positive (linear) correlation between the concentration of terpenoids (or antioxidants) and DPPH clearance activities. How about the (linear) correlation between the total triterpenoid level in the extracts and DPPH clearance activities?

Response 1: Thanks for your suggestion. According to the present results, the total triterpenoid level in the extracts and DPPH clearance activities is not completely linear. Total triterpenoid extracted from Ganoderma lucidum by ethanol might be content phenolic compounds and steroids, as well as have the antioxidant activity and resulted in nonlinear correlation between the concentration of total terpenoids and DPPH clearance activities [1]. So, in this study,we did not mention the correlation between terpenoids and DPPH clearance activity.

Point 2: Some terpenoids/steroids compounds could regulate/inhibit the level of NF-κB through the phosphorylation, DNA binding and p65 translocation. The related biomechanisms need to be investigated.

Response 2: The present results show that our triterpenoid-containing extracts could regulate/inhibit the level of NF-κB through the phosphorylation. Wang et al. [2] also used IHC to evaluate NF-κB expression in cardiac tissues. So we thought that the available data can illustrate the ethanol extract G42 can significantly improve NRF2/NF-κB-related oxidative stress and inflammatory response. The related mechanism investigation is our future work. And we have added discussion in the revised manuscript. Please refer to lines 314 to 317 in the revised manuscript.

Point 3: IHC results could not adequately reflect the expression level of NF-κB and other factors. Herein the correlation between expression level of NF-κB and other factors and triterpenoids/ extracts concentration need to be tested by western blot.

Response 3: We quantified the results of the IHC and supplemented in the manuscript,quantitative analysis results showed that the LG, MG, HG and PG could reduce the expression levels of the three proteins. These data are significant that can react to the correlation between expression level of NF-κB and other factors and triterpenoids/extracts. We have added a the result and change the Figure 5 in the revised manuscript. Please refer to lines 216 to 220 in the revised manuscript.

Point 4: How about the bioactivity-based MS molecular network? To understand the related bioactivities of triterpenoid-containing extracts, bioactivity-based cluster molecular network is better to be presented.

Response 4: Thanks for your suggestion. The bioactivity-based MS molecular network would be a fit-for-purpose method to further investigate bioactive candidate compounds in the ethanol extract of rice solid-state fermentation of Ganoderma tsugae. However, according to the workflow of bioactive molecular networking [3], the performance of an untargeted LC-MS/MS analysis and the evaluation of the bioactivity level for each chromatographic fraction are required to construct the bioactive molecular network. In this work, we just evaluated the antioxidant and hepatoprotective activity of the ethanol extract of G42. Chromatographic fractions of the extract were not prepared, while the antioxidant and hepatoprotective activity of each fraction were also not tested. Therefore, we cannot construct the bioactivity-based MS molecular network based on the current data. However, the reviewer provides a promising method to discover bioactive candidates in the extract. We plan to conduct these experiments in further study.

Point 5: Many triterpenoids have cytotoxicity under high concentration due to their cell/plasma membrane disruption effect. Herein, the apparent cytotoxicity of triterpenoids/extracts need to be evaluated by MTT or CCK-8 assays.

Response 5: We did not do cytotoxicity experiments, but we have conducted acute toxicological experiments in mice, recording the appearance (body posture, hair erection), clinical signs (breathing, caste, tremor, spasm) and the time of occurrence, remission and disappearance of each animal. After administration for 14 days, the mice did not die and other abnormal symptoms, the weight growth trend of the animals in the control group was normal and basically consistent, and no lesions occurred in the various organs of the mice.

Point 6: The action mechanism for the triterpenoids/extracts to the main bio-factors is better to be presented in a scheme or a table.

Response 6: Thanks for your suggestion. We found that column charts are a regular representation in journals to illustrate the results of bio-factors [4-6]. So we still use column charts to presented the result of bio-factors.

Point 7:  Commercially available triterpenoids (platycodigenin, cucurbitacin IIb and ganolecidic acid B) is better to be utilized as positive controls for the bio-evaluation of extracts.

Response 7: For the analysis of triterpenoids component in the ethanol extracts of G42 by HPLC-MS, it is better to select these three commercially available triterpenoids as the positive controls. But the biological evaluation made here is the evaluation of the hepatoprotective activity of the extract, three commercially available triterpenoids were not used as hepatoprotective drugs. So we considered it appropriate to choose a commercially available drug silymarin with hepatoprotective activity as a comparison.

Point 8: The conclusion part is not well-organized, the main results need to be presented in detail.

Response 8: We agree with the reviewer and thank you for your suggestions. We have revised our conclusions.

Point 9: There are some errors in the reference part, e.g., ref.19. “…Nutrients…” (year, volume and page information lost), ref.38. “Science and Technology of Food Industry 2019, 40, 53-57.” (abbreviation/full name) The authors need to carefully check and correct the reference part.

Response 9: According to the reference guideline, we have corrected the mistakes.

References

eljović S.; Veljović M,; Nikićević N.; Despotović S.; Radulović S.; Nikšić M.; Filipović L. Chemical composition, antiproliferative and antioxidant activity of differently processed Ganoderma lucidum ethanol extracts. J. Food Sci. Technol. 2017, 54, 1312-1320.

Wang, W.; Gu, H.; Lin, Y.; Yao, X.; Luo, W.; Lu, F.; Huang, S.; Shi, Y.; Huang, Z. SRC-3 Knockout Attenuates Myocardial Injury Induced by Chronic Intermittent Hypoxia in Mice. Oxid. Med. Cell Longev. 2021, 2021: 6372430.

Nothias, L.F.; Nothias-Esposito, M.; da Silva R.; Wang, M.; Protsyuk, I.; Zhang, Z.; Sarvepalli, A.; Leyssen, P.; et al. Bioactivity-Based Molecular Networking for the Discovery of Drug Leads in Natural Product Bioassay-Guided Fractionation. J. Nat. Prod. 201881, 758-767.

Yang, Z.; Liu, B.; Yang, L.E.; Zhang, C. Platycodigenin as potential drug candidate for Alzheimer's Disease via modulating microglial polarization and neurite Regeneration. Molecules 2019, 24, 3207. 

Yang, B.Y.; Cheng, Y.G; Liu, Y.; Tan, J.Y.; Guan, W.; Guo, S.; Kuang, H.X. Ameliorates Imiquimod-Induced Psoriasis-Like Dermatitis and Inhibits Inflammatory Cytokines Production through TLR7/8-MyD88-NF-κB-NLRP3 Inflammasome Pathway. Molecules 2019, 11: 2157.

Zhao, C.; Fan, J.L.; Liu, Y.Y.; Guo, W.L.; Cao, H.; Xiao, J.; Wang, Y.; Liu, B. Hepatoprotective activity of Ganoderma lucidum triterpenoids in alcohol-induced liver injury in mice, an iTRAQ-based proteomic analysis. Food Chem. 2019, 15, 148-156.

Reviewer 2 Report

Title: Hepatoprotective activity of ethanol extract of rice solid-state fermentation of Ganoderma tsugae against CCl4 -induced acute liver injury in mice

Herein, the authors develop, a new SSF method of G. tsugae using cooked rice as the sole solid substrate, and a wild strain of G. tsugae (G42) with high triterpenoid content was screened from nine strains. The fermentation method was optimized in terms of inoculation amount, fermentation temperature, and culture time by response surface methodology, to achieve high triterpenoid yield. The hepatoprotective activity of ethanol ex- 84 tract of G42 was evaluated by analyzing superoxide dismutase (SOD), glutathione peroxidasel (GSH), malondialdehyde (MDA), as well as inflammatory factors interleukin-2 (IL- 86 2), interleukin-8 (IL-8), interleukin-10 (IL-10), tumor necrosis factor-α (TNF-α), Interferon- 87 β (IFN-β) and interleukin-1β (IL-1β)) levels in a CCl4-induced acute liver injury mouse 88 model. The main triterpenoids of the ethanol extract were identified by high-resolution mass spectrometry and molecular networking

Major issues

1.       Structural activity relationship (SAR) of detected compounds with respect to the assumed activity should be discussed

2.       Molecular docking experiment should be done in silico

Minor issues

1.          The structures should be drawn using chemdraw or any other software

2.          The manuscript contains some spelling, grammatical and formatting mistakes that should be revised carefully

3.          The references should be carefully checked to be all in the same style.

Author Response

Response to Reviewer 2 Comments

Thank you for your comments. Our answers to your points are as follows.

Major issues

Point 1: Structural activity relationship (SAR) of detected compounds with respect to the assumed activity should be discussed

Response 1: Thanks for your suggestion. We agree that the structural activity relationship model can help reveal the potential effects of G42, and we have added a paragraph to discuss it in the revised manuscript. Please refer to lines 318 to 324 in the revised manuscript.

Point 2: Molecular docking experiment should be done in silico

Response 2: Molecular docking is a good approach for predicting targets for components in G42, thus helping to reveal the comprehensive effects of G42. However, the main purpose of this work is to screen a strain of G. tsugae and evaluate the hepatoprotective activity of the ethanol extract of rice solid-state fermentation of the strain. According to the result, we found G42 could produce high level triterpenoids, which also exhibited good hepatoprotective activity. Besides, by using mass spectrometry molecular networking, we investigated three main triterpenoids, namely platycodigenin, cucurbitacin IIb and ganolecidic acid B, which might responsible for the hepatoprotective activity. In further study, we will perform in silico molecular docking experiment of these compounds, and also conduct in vitro and in vivo assays to confirm their activities.

Minor issues

Point 1:  The structures should be drawn using chemdraw or any other software

Response 1: We draw the structures using chemdraw. Moreover, we have added the structures of the platycodigenin, cucurbitacin IIb and ganolecidic acid B in Fig 6C in the revised manuscript.

Point 2: The manuscript contains some spelling, grammatical and formatting mistakes that should be revised carefully 

Response 2: We have improved the overall English language of the manuscript and correct the spelling, grammatical and formatting mistakes.

Point 3: The references should be carefully checked to be all in the same style.

Response 3: According to the reference guideline, we have corrected the mistakes.

Reviewer 3 Report

The authors studied the ethanolic extract of a strain of the fungus Ganoderma tsugae, through solid state fermentation of rice (G42); antioxidant capacity by DPPH elimination rate (%), triterpenoid content, and chemical identification analyzed by OUPLC-Q-TOF/MS.

They analyzed the antioxidant, anti-inflammatory and hepatoprotective activity of the ethanolic extract of G42 in a model of liver damage induced by CCl4 in mice by the levels of GSH and MDA, and SOD enzyme activity; the hepatoprotective activity of serum ALT and AST activities and the expression of markers IL-10, IL-1B, IFN-B, TNF-a, IL-2, IL-8.

The ethanolic extract of G42 had anti-inflammatory, antioxidant, and hepatoprotective effects in CCl4 mice and identified three medically important triterpenoids: platycodigenin, cucurbitacin IIb, and ganolecidic acid B.

The conclusions show that this optimization of this fermentation method is effective in developing a food with a hepatoprotective effect, which is supported by the results reported.

The work is very interesting, the method of cultivating organisms that produce biocomponents of medical importance is important because their production is very slow.

However, the study uses very specialized languages that are not very easy to understand by the public.

Could you explain at the beginning of your results, the numbers of strains you used and the differences between them. Also, in their chromatograms they could point out the absorbance peaks of the compounds that they describe in their work.

Author Response

Response to Reviewer 3 Comments

Thank you for your comments. We have improved the overall English language of the manuscript. Our answers to your points are as follows.

Point 1: However, the study uses very specialized languages that are not very easy to understand by the public.

Response 1: In order to make it easier to understand by the public, the English expression has been modified in the manuscript.

Point 2: Could you explain at the beginning of your results, the numbers of strains you used and the differences between them. Also, in their chromatograms they could point out the absorbance peaks of the compounds that they describe in their work.

Response 2: As suggested, we have explained the differences between the numbers of strains. These 9 Strains of G. tsugae have genetic diversity, so further experiments were conducted on them. Please refer to lines 98 to 100 in the revised manuscript. We agree with the reviewer point out the absorbance peaks of the compounds,so we point out the absorbance peaks of platycodigenin, cucurbitacin IIb, and ganolucidic acid B that the triterpenoids only detected in the ethanol extract of G42 while not in rice.

Round 2

Reviewer 1 Report

The reviewer’s comments were addressed by the authors in the revised manuscript of molecules-1857010.R1.

Reviewer 2 Report

No additional comments